# Children, Media and Food. A New Paradigm in Food Advertising, Social Marketing and Happiness Management

**DOI:** 10.3390/ijerph18073588

**Published:** 2021-03-30

**Authors:** Rodrigo Elías Zambrano, Gloria Jiménez-Marín, Araceli Galiano-Coronil, Rafael Ravina-Ripoll

**Affiliations:** 1Audiovisual and Advertising Department, Faculty of Communication, University of Seville, 41012 Seville, Spain; rodrigoelias@us.es; 2Marketing and Communication Department, Faculty of Social Sciences and Communication and INDESS, University of Cádiz, 11406 Jerez de la Frontera, Spain; araceli.galiano@gm.uca.es; 3Business Organization Department and INDESS, Faculty of Economics and Business, University of Cádiz, 11406 Jerez de la Frontera, Spain; rafael.ravina@uca.es

**Keywords:** advertising, childhood, health, media, PAOS Code, television

## Abstract

The growing number of children who are obese or overweight in certain countries or geographical areas is a fact, as evidenced by the continuous studies and reports on the subject, endorsed or carried out by the World Health Organisation and independent research. In this context, food and beverage advertising can contribute to this. The main objective of this research is to evaluate compliance with the Food and Drink Advertising Code for Children (PAOS Code) in Spain and its relationship with nutritional habits on television, specifically on channels aimed at children. The methodology is therefore mixed: on the one hand, a qualitative technique based on discourse analysis and, on the other, a quantitative technique based on the content analysis of the advertising broadcast for seven consecutive days on three specialised channels and two generalist channels on Spanish television. The results reveal a systematic noncompliance with this code, which translates into inadequate eating habits among children. The immediate conclusion is that 9 out of 10 parts of food and drink advertising do not comply with any of the rules of the PAOS Code and that self-regulation by the advertising companies is negligible and insufficient.

## 1. Introduction

In recent years, obesity and overweight in children have increased dramatically [1]. This disease is considered a global epidemic [2,3,4]. It has significant consequences, such as psychiatric, psychological and social disorders in childhood [5,6], and an increased risk of developing noncommunicable diseases (hereafter NCDs) [7] throughout life, which will most likely lead to a socially complicated adult life [8], in addition to the difficulty of treating obesity, which is difficult in itself. It implies that overweight children are highly likely to become obese adults. It has led member states of the World Health Organization (WHO) to endorse the goal of no increase in childhood obesity by 2025 [9]. For more information, in 2016, more than 1.9 billion people over 18 were defined as overweight and more than 650 million as obese [10]. It shows that obesity has tripled in the last 40 years (between 1975 and 2016), which means that it has become a truly global epidemic of “Globesity” [11].

This phenomenon is even more worrying in the case of children. Estimates of obesity or overweight in developed or emerging countries for children aged 5 to 17 years put the problem at 22% of their population [12]. This data is 12 points higher than in developing countries [1]. In this respect, predictive models used by the UK Foresight Report [13] have suggested that 55% of the British population could be obese in the UK by 2050. In Spain’s case, the ALADINO [14] 2019 Study on Diet, Physical Activity, Child Development and Obesity in Spain reveals that 23.2% of children from families with fewer than 18,000 Euros gross per year suffer from obesity. According to the Spanish Health Ministry [15], the figure drops to 11.9% in families with an income of more than 30,000 Euros. Be that as it may, childhood obesity has tripled in Spain over the last two decades, precisely 11.1%. It has resulted in Spain having an alarming 16.1% of obese children. It has led Spain to occupy second place in Europe for the prevalence of childhood overweight in the 6−12 age group [16,17]. The media also play an essential role in this context. Moreover, in this respect, current legal regulations aimed at preventing childhood obesity are limited in their scope. This is due, on the one hand, to the fact that self-regulation is not effective [18,19], and on the other hand, that regulations concerning codes for the dissemination of images in the media and the promotion of other types of activities, such as the promotion of a healthy lifestyle, are relatively lax and ineffective [20,21,22].

In this sense, it can be affirmed that the media, in general, and advertising, may be factors responsible for promoting unhealthy diets and lifestyles [23,24]. Thus, several organizations and groups have spoken out to promote measures to ensure that minors are genuinely protected against advertisers’ abusive campaigns. Examples include adopting a system of bylaws that uses a nutritional profile model to restrict the display of unhealthy products or the imposition of a legal ban on any food and drink advertising aimed at minors [25].

There is a wide range of ultra-processed foods characterized by aggressive advertising in the Spanish spatial framework, mainly aimed at children [26], who are very vulnerable to advertising stimuli of a negative nature [27]. The literature shows that children who consume less children’s advertising enjoy a healthier life, and therefore greater subjective happiness [28,29,30,31]. It should be noted that this age group is characterized by low cognitive resources to decode the messages correctly received [27,28,29,30,31].

As is well known, one of the most critical stakeholders influencing children’s eating behavior is parents. In this regard, Grier et al. [32] show that parents’ greater exposure to fast-food marketing is associated with more frequent consumption of fast food by their children. It is also contributed to some advertisements for processed and ultra-processed products that communicate to potential customers that their products are healthy [33].

At this point in the introduction, it is worth noting that two main factors influence childhood obesity. The first is that food advertising on TV increases energy-dense foods aimed at children, significantly overweight [34,35]. Secondly, overweight children consume more food from prestigious, globally known brands [36]. Given the above, it is not surprising that many food multinationals currently set their target market on children. Behind this economic practice, we can see a lack of business ethics aimed at protecting the responsible consumption of the youngest household members and a lack of legislative regulation to encourage children to consume healthy food for their bodies [37]. In this way, what certain food companies are doing is educating children on the consumption of their products so that, as later young and adult people, they will have these patterns and trends in food and beverage purchasing and consumption behaviour [38].

### Social Marketing, Happiness Management and Advertising for Healthy Eating in Children

In food, the application of social marketing goes beyond informing about the value of food (nutrients, health benefits and interest at certain ages). Food is related to tradition, society’s values, family structure and the roles played by its members [39]. For this reason, in social marketing, it is common to use advertising to contextualize food consumption in the group’s culture and try to generate a positive attitude towards the food being promoted [40].

The culture of happiness management, together with social marketing, can help make a Copernican turn in the healthy habits of the youngest members of 21st-century households. To achieve this goal, companies must be aware that today’s society demands that organizations implement business strategies to maximize customer satisfaction in consuming their food products [41]. The literature shows, then, that this fact no longer comes only from offering low prices but also from responding to the nutritional needs of their consumers, including providing products or foods without the use of exogenous chemicals [42].

Companies that carry out economic policies to achieve this end will be characterized by understanding their potential customers’ healthy wellbeing as a strategic, sustainable and competitive factor [43]. Under this umbrella, companies’ corporate governance will enjoy a corporate image based on the healthy nutrition (subjective) happiness binomial. As is well known, this construct is an excellent instrument to show the market that their corporate culture is oriented towards the health of their consumers and, primarily, towards the health of the youngest [44]. However, producers of healthy foods may need policymakers and other health promoters and prescribers to effectively market their products [45]. In this sense, companies should engage in social marketing initiatives with positive and emotional messages to encourage parents to include fruit and vegetable consumption in their children’s diets in the Covid-19 era [46]. They must be aware that public authorities have an urgent moral obligation to regulate self-breaches of certain ethical norms and take legal action against companies that are unfair to consumers and, therefore, to their nutrition health.

This article analyzes food and beverage advertising on mobile devices aimed at children. This study found that many of the food products are bought or ordered as a direct result of this advertising.

## 2. Hypotheses and Objectives

The objectives of this research are based on the need to expose a situation related to food and its advertising focused on a public such as children, always aiming to improve public health. So, the general and specific objectives are:-Determine what ethical principles and standards food advertisers do or do not violate in their television advertising.-Contrast the advertising rigor of the PAOS Code [47] with the advertising actions carried out by agencies within the Spanish scope of action. In this way, it will be possible to identify the main advertising techniques and strategies used by brands in the Covid-19 era.-Assess the existence of professional ethics in this advertising sector.

If we assume that unethical use of advertising aimed at children can harm the health of the most vulnerable people, and in accordance with the literature reviewed, we start from the following conjectures (hypotheses):Unhealthy “food and beverage” advertising promotes unhealthy eating habits for individuals and encourages the abusive consumption of products that are harmful to children, resulting in negative effects on the well-being of society as a group and detrimental to public health.Children’s advertising is not adapted to the precise language that messages aimed at children up to the age of 12 should convey. This leads to confusion among minors, which can result in unhealthy food and beverage products advertising that can lead to eating disorders or poor eating habits in children, such as overweight or obesity.

## 3. Materials and Methods

### 3.1. Presentation of the Case

Since 2005, the AESAN (Spanish Agency for Food Safety and Nutrition) has been developing the PAOS Code, i.e., the code for the Co-regulation of Food and Drink Advertising aimed at Children, Obesity Prevention and Health [47]. This code of ethics aims to promote physical wellbeing and holistic and subjective happiness among children through a healthy diet in today’s digital society. To this end, the leading television operators in Spain undertake to require that food and beverage advertisements aimed at children younger than 12 comply with all the standards set out in the code.

### 3.2. Material Design

Based on the secondary sources consulted for this article’s development, the authors of this research decided to use a descriptive hermeneutic methodology. Once this question was resolved, a cross-sectional study of food and drink advertising aimed at children under 12 years of age on Spanish television was carried out. To this end, we examined the children’s advertisements broadcast on the Disney Channel, Neox, Boing, Telecinco and Antena 3 between 14 and 20 December 2020. The choice of these channels is justified, on the one hand, by the fact that Disney Channel and Boing occupy the highest positions in the ranking of channels whose programming is aimed exclusively at children. On the other hand, Neox, Telecinco and Antena 3 have the highest market shares in the current panorama [48].

In conjunction with the above, it should be noted that the advertisements analyzed were made in the time slots between 5 pm and 8 pm (in the case of working days) and between 9 am and 12 noon on Saturdays, Sundays and public holidays. The choice of this time slot is basically because Law 7/2010, of 31 March, the General Law on Audiovisual Communication [49] establishes that in this time frame of reference, it is prohibited to broadcast audiovisual content that harms the physical, mental or moral development of minors under thirteen years of age. Once this assessment has been made, it can be noted that the sample population obtained was 177 advertisements.

The reason for using the hermeneutic approach is due to the observation of this same methodology in similar investigations applied to other media or in other geographical areas, but with the same purpose of study [34,41,45,46], and always as a way of providing data that help to safeguard public health.

In relation to measurement, this scale has been chosen because in the specific case of Spain we find ourselves with an age range which, despite being very different from one another on a psychological and sociological level, is considered the same on a legislative level (in terms of protection), in addition to the fact that the orientation of audiovisual content contemplates a scale of parameters ranging from 5 to 12 years of age.

### 3.3. Procedure and Data Analysis

To quantitatively explore whether the selected advertising messages were harmful to children’s health, the PAOS Code was used. It sets out 14 ethical standards (broken down into 32 specific standards). These fourteen standards are I. Principle of legality; II. Principle of loyalty; III. Nutrition education and information; IV. Product presentation; V. Product information; VI. Sales pressure; VII. Support and promotion through characters and programmers; VIII. Advertising Identification; IX. Comparative presentations; X. Promotions, sweepstakes, contests and kids’ clubs; XI. Security; XII. Processing of personal data; XIII. Viral marketing; and XIV. Protection against inappropriate content.

According to compliance with the 14 ethical norms and 32 ethical standards of the PAOS Code, our study population was classified by six researchers of recognised prestige (3 principal investigators, supervisors; 3 secondary investigators, support). This was done at three different levels: compliant, noncompliant and uncertain, based on the standards established by Leon et al. [50]. Based on this classification and according to its visualisation by the research team, the sample was typified. These data were then tabulated in an Excel table according to the following dimensions: total number of spots broadcast, television channels (generalist and specialised), types of products, product categories and duration of the spot.

The last point to consider about the methodology is that this research focuses exclusively on examining commercial messages aimed at children under 12 years of age. In this sense, there is a blurred boundary between 12 and 15 years of age where minors tend to watch television programs aimed more at adulthood or adolescence, while up to 12 years of age, the television programs watched by this population are child-oriented [51]. In order to estimate whether advertisements are aimed at this group, the authors of this scientific study developed the following three criteria:Food products are objectively promoted primarily to the public under 12 years of age.Advertising is designed so that, by its content, language, and images, it objectively targets children up to 12 years of age.Advertising broadcast in television programmers which, objectively, are aimed at children less than 12 years of age.

This will allow us to analyze the following aspects: the total number of spots aired, TV channels (generalist and specialised), types of products, product categories and duration of the spot.

## 4. Results

Throughout the time frame analyzed in this study, 326 food and drink advertisements were found broadcast on the channels selected by the authors of this research. Of these advertisements, 177 were aimed at children less than 12 years of age, which means that our study’s sample population is made up of 177 advertisements. Based on this information, a general description of the variables’ frequencies examined in our sample is shown in Table 1.

The table above shows four essential aspects: 67.2% of the adverts analyzed were broadcast on school days; boing is the TV channel with the most food and drinks advertisements aimed at girls (42.94%); 70.62% of the products advertised are food, 83.05% of which refer to products belonging to the nonessential food category; 55.93% of the commercial spots examined have in common that their duration is less than 21.7 s.

Having explained this frequency analysis briefly, we proceed to explore, on the one hand, the standard ethical principles that the elements in our sample comply with, and on the other hand, to quantitatively examine the specific rules of the PAOS Code that the 177 advertising pieces analyzed to comply with (Table 2). In this regard, the ethical standard with the highest rate of noncompliance with the PAOS Code is principle number V (Product information), with a value of 6.78%. This is because the communicative language used in these advertising messages is not suitable for children. However, the visual stimuli used are not, as they constitute a factor of an eye-catching nature that incites impulsive consumption by children under 12 years of age. It is also worth mentioning that another of the ethical standards with the lowest compliance rate concerning the code is standard IV. (Presentation of products), with a record of 10.17%. This is since the advertising pieces studied are characterized by the fact that they induce our target audience to misjudgements about the benefits of using the product. This phenomenon is caused by the fact that these commercials use fantastic elements (animations or cartoons) that exploit the images of children less than 12 years of age, without realizing that they generate unattainable expectations, which violates their childish innocence.

Table 2 also clearly shows that standard X (Promotions, prize draws, competitions and children’s clubs) shows a high noncompliance with the PAOS Code, namely 58.19%. Behind this figure lies the fact that the advertising pieces examined recurrently violated the ethical standards concerning the advertising message, promotional offers, advertising prizes and references to children’s clubs.

Following the previous paragraph, we cannot fail to note that this is also the case in ethical standards XI (Safety), with a record of 43.5% noncompliance with the PAOS Code. Therefore, it is not surprising that the commercials examined in this study violate the ethical standards linked to the prohibition of images or messages that encourage the dangerous or inappropriate use of the advertised product and advertisements that encourage children under 15 to enter strange places or converse with strangers [47].

Based on what has been read under this heading, it is not trivial to point out that the advertising messages observed continuously violated the 14 ethical standards of the PAOS Code. Compliance with which is voluntary for companies in the food and beverage sector. Focusing on this phenomenon, it is also interesting to note that the advertising spots observed in this research regularly contravene ethical standards IV (Product presentation), X (Promotions, prize draws, competitions and children’s clubs) and XI (Safety) of the PAOS Code. See Figure 1.

Going a little deeper into the quantitative analysis of this issue, it is worth noting that the commercial pieces examined work against a large percentage of the 32 ethical standards that make up the normative corpus of the PAOS Code. This is particularly evident in the ethical standards relating to broadcasting, online advertisements, the communicative language of advertising spots, the benefits of product consumption and commercial promotions (Figure 2).

Given the above, it is pertinent to know which products in our sample population mostly fail to comply with the ethical standards of the PAOS Code. The following Table 3 was designed for this purpose.

This shows that dairy advertisements breach ten ethical standards, i.e., 31.25% of the 32 standards that make up the PAOS Code, followed by bakery and biscuit advertisements with a level of noncompliance of 25%. In contrast to these high values, advertising messages for sugary drinks, sauces, and snacks have a low level of noncompliance, namely 6.25%. At this point, it is worth noting that the ethical standards most frequently breached by the products in our study are numbers 4 (dissemination of advertisements on the Internet), 9 (language understandable to minors), and 19 (advertising of promotional offers). This occurs in exactly five products and brands, representing 45.45% of the total number of products analyzed in this study. Precisely, these advertisements correspond to Kinder Sorpresa, Chocapic, Nesquik, Haribo and Choco Flakes.

Given this wealth of information, the following is a detailed description of the advertisements that infringe the PAOS Code based on the following dimensions: days analyzed, television channels, types of products, product categories and time duration of the spots (see Table 4). In this regard, it should be noted that the following five findings are incredibly striking. The first is that 57.1% of noncompliance occurs on weekdays (Monday–Friday). Secondly, 82.84% of the advertisements analyzed are broadcast on specialised channels that regularly violate the PAOS Code. Thirdly, 60.5% of the food advertisements examined by this research contravene the code we are examining. Fourthly, 72.9% of the advertising pieces relating to nonessential products do not comply with the analysis regulations. Moreover, lastly, 55.9% of the spots considered in this study with a duration of fewer than seven seconds do not comply with the PAOS Code.

## 5. Discussion

Based on data explored in this research, it can be concluded that around 90% of the analyzed pieces do not meet any of the standards of the PAOS Code. It should be noted that the standards with the highest levels of incompatibility with this code are those that attempt to safeguard the vulnerability and innocence of children (a fact that takes on relevance when it comes to their nutrition).

Currently, advertising agencies producing food spots for children are characterised by continuous and consistent noncompliance with the PAOS Code. Many industrial products’ common denominator is that they are highly detrimental to children’s health and, thus, their subjective wellbeing. They do not contribute to reducing obesity and overweight in children. This is undoubtedly one of the biggest social problems facing the top leaders of public health in advanced countries, including Spain. It is appropriate to point out that in 2020 the Spanish Sports Council will carry out the ADB Plan 2020 to awakening interest in sport among young people and children [56]. Those responsible for its communication strategy thought that the ADB Plan 2020 could enjoy greater global visibility with the participation in its spots of prestigious brands in the children’s food sector.

Companies offering products with high sugar content were involved. According to several studies [57,58,59,60], the added sugars’ mission is to increase food and preservation palatability. Therefore, the ingestion of these ultra-processed foods with a zero nutritional profile is contrary to a culture of sport and health that the Plan ADB 2020 promotes in the Spanish youth population. This is no trivial matter when the WHO [61] points out that the rate of childhood obesity will exceed that of underweight by 2022. This shows that the supply of ultra-processed products and eating habits contributes to maintaining the current emerging obesogenic society.

For this reason, now more than ever, it is essential to undertake social marketing strategies aimed at curbing the constant growth of child globesity. At present, one of the techniques most used by advertising agencies is to design advertisements that generate involuntary brain stimuli towards the ingestion of hypercaloric and ultra-processed foods in small children, both average weight and obese [62]. In a television ecosystem where 33.2% of broadcast food advertisements are positively associated with high-calorie, energy-dense, low-level food choices [63].

Perhaps nutrition education at an early age helps children acquire healthy eating habits that improve their quality of life. This way, it can be combated with advertising pieces that gravitate their corporate marketing through minors’ vulnerability. A practice that only seeks to maximise economic benefits without considering the harmful effects these types of food have on future adults’ health in Spanish society. This communication (and marketing activity) is supported by the absence in the PAOS Code of specific rules concerning the nondeclaration of interest conflicts.

One of the most important findings of this research is that only one of the 177 advertising pieces examined complies with the PAOS Code. This corresponds to the Suchard Christmas campaign spot. We would not like to end this section without acknowledging the existence of some limitations of our study. One of them is subjectivity, which is due to the complexity of some of the ethical standards analyzed. In this sense, to reduce this bias error, the research’s reliability was increased through our six specialists’ cognitive consensus. Another limitation is the lack of representativeness of the sample and the time frame of the data collection. Despite these two drawbacks, we believe that this descriptive study’s object is quite representative, which may help future research assess the degree of relevance that the mass broadcasting of advertisements for food products that do not comply with the PAOS Code have on the health of Spanish children.

## 6. Conclusions

This article has sought to highlight the urgent need for public authorities to undertake communication and marketing actions aimed at highlighting the harmful consequences of poor eating habits for children so that current and future parents in the advanced world can be made vitally aware of the fact that childhood obesity is one of the significant public health problems. The latest study by the World Health Organization’s European Childhood Obesity Surveillance Initiative [64] quantifies that around 40% of the European Union’s child population is obese or overweight. As pointed out in previous sections, this is mainly since our children’s diets are based on the ingestion of a high percentage of ultra-processed foods, precisely 30% [65]. It is therefore not surprising that the literature defines childhood obesity as the epidemic of the 21st century. Such a definition will be difficult to eliminate in the short and medium-term. This assertion is supported by the fact that, in Spain, around 50% of commercial spots for drinks and food aimed at children under 12 advertise unhealthy products [66]. This reflects the asymmetry between the commercial objectives of the ample food and beverage companies that broadcast these advertisements and the solution to the nutritional crisis affecting children born after 2011 [67]. From this perspective, this advertising can be described as misleading and harmful to children’s health.

It is necessary to point out that noncompliance with the PAOS Code is more common on specialised channels than on general channels. This is due basically because they broadcast a higher volume of programs, which infringe the innocence of children under 12 years of age. One of the reasons that may explain this is to be found in the lax self-regulation of the advertising companies that design advertisements aimed at minors. Hence, this group is the primary victim of their commercials. It is therefore urgent that legislative measures be put in place to protect the integrity of children. From our point of view, this can be achieved through two necessary actions. The first is that the current legislation should cease to be voluntary. The second is that future administrative regulations should be aimed exclusively at all agencies and advertisers targeting minors.

The scientific evidence shows a food industry that incites those responsible for public health bodies and the political class to promote productivity growth that is antithetical to current public health regulations and, therefore, to its citizens’ subjective happiness [68]. An excellent example of the pressures exerted by giant food corporations at the advertising level is found in Spain, with a voluntary code of co-regulation of food and drink advertising aimed at minors, dating from 2005, ignoring international organisations’ recommendations, such as those of the WHO. It is therefore not surprising, as this research has shown, that most of the products advertised aimed at innocent children in this country, show products with a high nutritional value, which do not correspond to reality: they are, in practice, food products with low (or no) nutritional value and high in calories. This leads to the public administration’s imminent need to regulate, in a compulsory manner, legislative laws, such as tax increases, for example, that, in addition, acting in favour of healthier consumption [69], put an end to these advertising claims that falsify the quality of the product as a guarantor of health. It will then be possible, on the one hand, to restrict the massive overexposure of children to unhealthy products or products with zero nutritional profiles and, on the other hand, to question the actions of each of the agents involved in this process in order to find out whether their desire for profit or the code of ethics of their profession prevails.

In our view, happiness management and social marketing can play a vital role not only in making organisations aware of the need to develop advertising campaigns that cultivate the consumption of healthy products as a vehicle for personal and subjective happiness but also in encouraging parents to encourage their children to lead a healthy life, based on sport and the consumption of foods low in calories and saturated fats. In conjunction with the above, it would be exciting for the Ministry of Consumer Affairs to undertake preventive health programmed in collaboration with companies to stimulate healthy product consumption. This programming should include an explicit ban on misleading food and beverages for both children and general consumers.

Through these arguments, and by way of conclusion and practical implication, it would be interesting for happiness management and social marketing specialists to promote the creation of business certificates to recognise those foods or products (both individually and jointly) that significantly enhance the subjective healthy happiness of their consumers. Under this approach, a group of researchers from the University of Cadiz have registered the *Biofelicidad* brand at the Spanish Patent and Trademark Office of the Spanish Ministry of Industry, Trade and Tourism (Figure 3). This brand represents a new perspective and addition to the fast food sector, demonstrating that it is possible to create productive, healthy and wholesome links. This constitutes a DNA of progress, collective wellbeing, sustainability and happiness for the new generations [20], based on Pablo Picasso’s famous phrase: “Your best capital is your health”.

## Figures and Tables

**Figure 1 ijerph-18-03588-f001:**
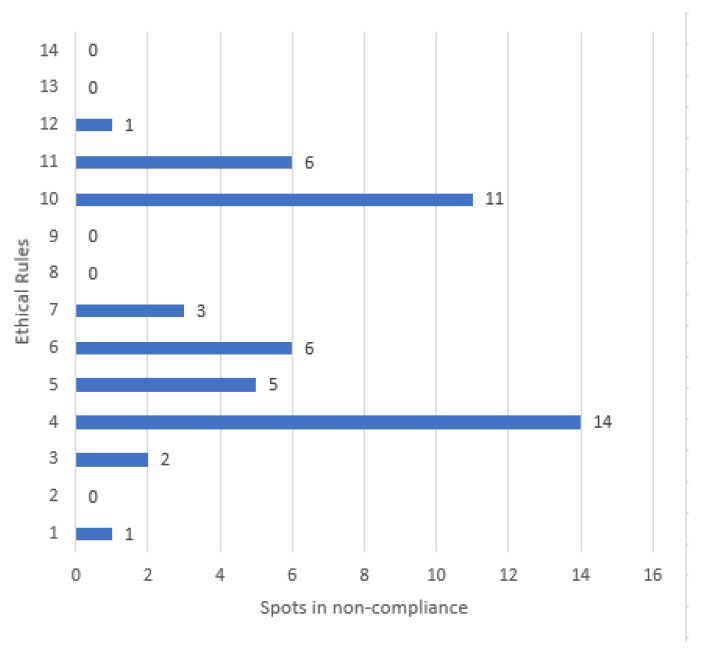
Noncompliance with general ethical rules in Food & Beverage Ads (FBA) in Spanish TV in 2020.

**Figure 2 ijerph-18-03588-f002:**
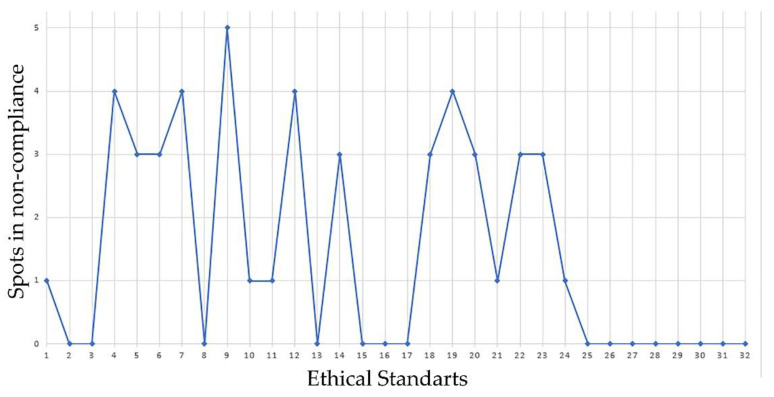
Noncompliance with the general ethical standard in FBA in Spanish TV in 2020.

**Figure 3 ijerph-18-03588-f003:**
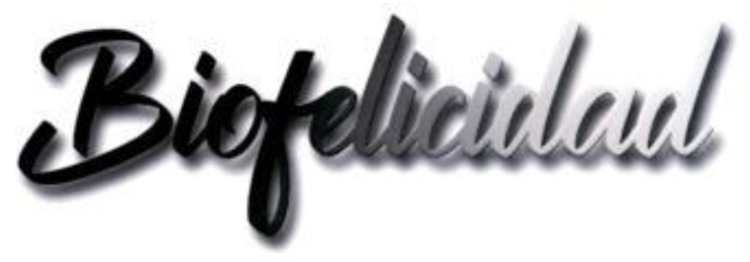
*Biofelicidad* brand.

**Table 1 ijerph-18-03588-t001:** Frequency of food and drink advertising for the under-12s in our sample population; by days of the week and TV channels, type of products, product categories and duration of the spot.

Analysed Items	*n*	%
Analyzed days		
Monday to Friday	119	67.20
Saturday and Sunday	58	32.80
Total	177	100
Television channels		
Generalists (*n* = 25)		
Telecinco	12	6.78
Antena 3	13	7.35
Specialized (*n* = 152)		
Boing	76	42.94
Neox	39	22.03
Disney Channel	37	20.90
Total	177	100
Types of products ^1^		
Food	125	70.62
Beverage	52	29.38
Total	177	100
Product categories ^2^		
Essential	0	0
Nonessential	147	83.05
Miscellaneous	30	16.95
Total	177	100
Length of spot		
Less than 21.7 s	99	55.93
More than 21.7 s	78	44.07
Total	177	100

^1^ In order to provide readers with a better understanding of the types of products analyzed, it should be specified, on the one hand, that the products classified as “food” in this study are made up of chocolate, pastries, savoury snacks, bread, rice, dairy products, cereals, cold meats, pizzas, sauces, dressings, prepared broths and snacks. On the other hand, “drinks” are dairy products, sugary drinks, gazpacho and water. ^2^ This research is based on the definition of: “Essential products”: those dietary products of frequent use that support the physical and mental development of the child. “Nonessential products”: those products that have low, poor or no nutritional value. “Miscellaneous”: those products that are not strictly essential for human development and have an average nutritional value and whose moderate intake is not considered harmful [52,53,54,55].

**Table 2 ijerph-18-03588-t002:** Compliance with the Drink Advertising Code for Children (PAOS Code) and its ethical rules in Spain, 2020.

Ethical Rules	Compliant		Noncompliant		Uncertain	
	*n*		*n*		*n*	
I. Principle of legality	85	48.02	51	28.81	92	51.98
II. Principle of loyalty		100	-	-	-	-
III. Nutrition education and information		40.68		20.34	69	
IV. Presentation of products		10.17	65	36.72		55.37
V. Product information		6.78		41.8	100	56.5
VI. Sales pressure		30.5	71	40.11	52	29.38
VII. Support and promotion through characters and programmers		84.18		15.82	-	-
VIII. Identification of advertising		100	-	-	-	-
IX. Corporate presentations		100	-	-	-	-
X. Promotions, prize draws, competitions and children’s clubs		41.81	103	58.19	-	-
XI. Security	63	35.59	77	43.5		20.9
XII. Processing of personal data	148	83.62			29	16.38
XIII. Viral Marketing		87	-	-		
XIV. Protection against third parties for inappropriate content		87	-	-		

**Table 3 ijerph-18-03588-t003:** Frequency of noncompliance ethical standards of the PAOS Code by product category.

Product Category	1	4	5	6	7	9	10	11	12	14	18	19	20	21	22	23	24	Total
Chocolates and confectionery products																		6
Pastries and biscuits																		8
Savory snacks																		4
Bread and rice																		5
Dairy																		10
Cereals																		3
Meat sausages and pizzas																		6
Sugary drinks and gazpachos																		2
Sauces, dressings and prepared broths																		2
Snacks and dietary substitutes																		2
Water																		4
**Total**	2	5	3	3	4	5	2	1	4	3	3	5	2	1	3	3	1	50

**Table 4 ijerph-18-03588-t004:** Overall noncompliance with the drink and food PAOS Code advertising to the under 12s in our sample by days of the week and TV channels, type of products, product categories and duration of the spot.

Days Analyzed	*n*	%
From Monday to Friday	101	57.1
Saturday and Sunday	58	32.7
Total	159	89.8
Television channels		
Generalists (*n* = 25)		
Telecinco	7	3.9
Antena 3	6	3.4
Total	13	7.3
Specialized (*n* = 152)		
Boing	76	42.9
Neox	33	18.6
Disney Channel	37	20.9
Total	146	82.4
Types of products		
Food	107	60.5
Drink	52	29.3
Total	159	89.8
Product Categories		
Essential	0	0
Nonessential	129	72.9
Diverse	30	16.9
Total	159	89.8
Duration of the spot		
Less than 7 s	99	55.9
Over 21.7 s	60	33.9
Total	159	89.8

## Data Availability

No new data were created or analyzed in this study. Data sharing is not applicable to this article.

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
