# Peer review of "Children, Media and Food. A New Paradigm in Food Advertising, Social Marketing and Happiness Management"

_ijerph, 2021, doi:10.3390/ijerph18073588_

Round 1

Reviewer 1 Report

It is an interesting investigation with shocking results. I hope that this effort contributes to developing initiatives that protect the health of children and adolescents in your country.

My observations are as follows:

I believe the word "figure" in the  first paragraph of the second page is not appropriate.

The wording of the hypotheses must be improved, in this case it must show cause and effect. Do not use period and followed.

The objectives are written in the infinitive verb.

You should be clearer with the explanation of the methodology. Why did they use a hermeneutical approach? Did the instrument have a measurement scale?

Write a paragraph after Table 4, each Table and Figure must be commented.

It is not usual, in a scientific investigation, to close the article with a Figure.

Author Response

Thank you very much for your comments. We will go into detail on each of them, once we have revised the manuscript according to your suggestions for improvement.

Thank you for helping us to improve:

  1. The word 'figure' was certainly not the most appropriate word. We have changed it to 'data'.
  2. The wording of the hypotheses has been improved by stating a cause-effect and not using a full stop.
  3. In relation to the methodology, the reason for using the hermeneutic approach has been explained as well as the measurement scale.
  4. Each table and figure have been commented on.
  5. It is true that it is not usual for a scientific research to close with a figure. However, we must clarify that this research work has resulted in the creation, registration and patenting of a brand that is directly transferred to society as part of the transfer of knowledge from the university to the citizens.

Thank you so much, again for every advice.

Reviewer 2 Report

I consider it to be an article with an interesting topic, but with an important weak point: its statistical analysis is very simple. I understand that the data they have through the methodology does not allow us to relate concepts or draw conclusions, but the analysis is superficial.

Revising each section, the review of the bibliography is extensive and deep and allows the reader to know the current situation of this problem.

However, I think that the objective and the hypothesis are changed (what the authors call objectives are hypotheses, and what they write as hypotheses are objectives).

Section 3 is lowercase, and perhaps it should make more sense if section 4 is incorporated into it.

The results cannot confirm and refute the hypotheses. Probably because the hypotheses are as deep as the objectives. A possible solution would be to rewrite the objectives and hypotheses so that they are more concrete and can be refuted or confirmed with the analyses carried out.

Once the objectives and hypotheses have been rewritten, the discussion should be rewritten and the conclusions of the study expanded.

Author Response

Thank you very much for your considerations and suggestions for improvement.

We will now respond to each of your comments:

  1. The strength of this research is not so much the quantitative statistical analysis as the qualitative analysis. We are aware that many researchers do not like qualitative methodology, but in the social sciences it works very well and yields real conclusions that can help, as in this case, to improve public health. In fact, the International Journal of Environmental Research and Public Health itself recently published a study (Food and Beverage Advertising Aimed at Spanish Children Issued through Mobile Devices: A Study from a Social Marketing and Happiness Management Perspective) where this same methodology was used. This article is being used by Autocontrol de la Publicidad (the organisation that regulates Spanish advertising legislation and ethical principles) to limit and control food and beverage advertising (as well as other sectors, such as video games) in pursuit of a greater state of well-being, happiness and health among Spaniards: current children (and their parents), future adolescents and future adults.
  2. Related to the bibliography, we have certainly tried to cover, within the limits of sanity, the most recent and related literature review without going overboard. Thank you for pointing this out.
  3. Regarding the objectives and hypotheses, they have been revised and adapted as stated by the reviewer and rewritten. Thank you very much.
  4. We have certainly followed the recommendations to add the sub-section 'Presentation of the case' under the methodology heading.
  5. The discussion has also been rewritten, expanding on the conclusions of the study.

Thank you so much, again. 

Reviewer 3 Report

Congratulations to the authors for the article, it is an interesting and novel article that can provide interesting conclusions for the improvement of the object of study and in this case with great applicability.
The abstract should argue the methodology, results and conclusion to a greater extent and not so much the introduction.
In the second paragraph of the discussion, apart from a summary of the main results, a series of opinions of the authors are offered, which I believe should be the result of the analysis of the discussion, and should therefore be found at the end of the discussion or in the conclusions.
The conclusion is confusing and there is no clear relationship with the conclusion offered in the abstract.
Reference 56 in the text is incorrect and reads [56ADB].
The discussion is somewhat sparse and does not make use of the references argued in the introduction and from which the hypotheses and objectives are assumed to stem.... This should be improved.
Revise the citations, reference 53 appears on page 11 for the first time and should be 62.

Author Response

Thank you very much for all your comments. We proceed to explain the improvements according to your suggestions:

  1. In the abstract, the methodology, the results and the conclusion have been argued and not so much the introduction.
  2. The discussion has been rephrased so that it is the result of the analysis.
  3. The conclusions have been rewritten to make them less confusing and to establish direct links with the results.
  4. Reference 56 has been corrected.
  5. The discussion has been improved as requested by the reviewer.
  6. The citations and all references have been revised.

Thanks again for all the feedback and comments for improvement.

Reviewer 4 Report

Thank you for the opportunity to review the manuscript entitled, "Children, Media and Food. A New Paradigm in Food Advertising, Social Marketing and Happiness Management”.

I believe this study investigated a topic relevant to the readers of “IJERPH”. My opinion is that in the current form the manuscript does not have scientific quality to be published in IJERPH. The manuscript is far below scientific standards.

The authors propose the following hypothesis “The advertising of unhealthy foods and beverages promotes unhealthy eating habits for individuals and encourages the abusive consumption of harmful products to children. This practice has adverse effects on the wellbeing of society”.   However, this study analyzes only one variable: "The advertising of unhealthy foods and beverages".

“Based on the secondary sources consulted for this article's development, the authors of this research decided to use a descriptive hermeneutic methodology”. Descriptive methodology is not enough. The descriptive studies do not study relationships between variables. Descriptive research does not answer questions about why a certain phenomenon occurs or what the causes are.

Finally, the manuscript does not meet my expectations for publication in the journal.

Thank you very much. 

Author Response

Thank you very much for your comments. We will now respond to each of them:

  1. The authors studied the food and drink advertising broadcast on the selected channels to see whether the ethical standards and principles embodied in the PAOS Code are met. It is not that the research analyses only one variable, but that it is the only one that has been found. It is true that it would be interesting to broaden the sample for the future and as a possible line of research. This is what we have noted following the reviewer's observation. Thank you very much.
  2. We do not use only a descriptive methodology. A combination of qualitative-quantitative methodology is used (discourse analysis + content analysis), a mix widely used in the social sciences and which, from the point of view of communication, generates good results. In fact, the methodology employed is like that followed by articles previously published by the International Journal of Environmental Research and Public Health, as well as other studies used as a source for the theoretical framework of the article.

In any case, certain parts of the article have been reworked and rewritten to shed more light on issues that have not been fully clarified for the reviewer.

Thank you, sincerely, for all the annotations and feedback.

Reviewer 5 Report

I read this manuscript with interest. It addresses an important topic and it is geneally well-written. However, I have some suggestions and considerations that could help clarify the message of the paper, as follows:

  • the first, and main, concen I have about the paper is that it appears to use a (too-simple) model to explain the links between food, health and happiness. The framework the authors use (as I take it) is that children and early-adolescents are exposed to unethical messages; these messages (negatively) influence their eating habits; these negative eating habits impact their health; their poor health is associated with low happiness. Now, I agree that this cycle is sensible. However, I think it is too simplistic. First of all, the concept of happyness can be subjective and the authors should define it very clearly based on literature (specifically in youths). Second, the authors assume that the unethical adverstisings are actually watched by youths. This is highly probable but not sure (youths could therefore not at all be exposed to those messages) (of course, they are very probably exposed, but this should be proven; it cannot be simply assumed). Thirdly, many other factors influence the above cycle, and this brings me to the second main concern about the paper:
  • the authors only briefly mention parents. This is a key point, in my view. Not only parents are models to their children with regards to eating habits. They are very frequently those who provide food to children and can influence youths' assumption of food through parenting styles. Not to mention parents who suffer eating disorders, who can pass the psychological difficulty intergenerationally. These considerations are completely absent from the paper and should be included (at least in a brief passage).
  • In the paper no difference is made based on age. Are 5 years old children, in terms of functioning, the same as 12 years old early adolescents? One should think not. No developmentally-oriented consideration appears in the paper, while I think it should.
  • Finally, although acknowledging that the authors rely on previous literature, I am concerned about the syllogisms "physical health=happiness" and "un-healthy=un-ethical". This sort of reasoning may serve as an unwanted support to "ethical protocols", while I would rather support "scientific protocols".  Therefore, I would endorse more prudent language such as "subjective happiness" (rather than "happyness" that could sound as an objective state, which is not).

Author Response

Thank you very much for all the suggestions and considerations raised. We will now respond to each one of them, trying to adapt to everything that has been raised:

  1. The relationship between food, health and happiness has been discussed in more than one article and by more than one author. It is true that the concept of happiness can be subjective. In this sense, many authors use the term "happiness" in their research as a synonym for subjective well-being. Here are some cases:
  • Seligman, M. (2016). Flourishing: The new positive psychology and the search for well-being. Mexico: Editorial Océano.
  • Ackerron, G. (2012). Happiness economics from 35000 feet. Journal of Economic Surveys, 26(4), 705-735. http://dx.doi.org/10.1111/j.1467-6419.2010.00672.x
  • Diener, E. (2000) Subjective well-being: The science of happiness and a proposal for a national index. American Psychologist, 55(1), 34-43. http://dx.doi.org/10.1037/0003-066X.55.1.34

  1. In the study, the authors do not claim that unethical advertisements are actually seen by young people, but during the selection of the sample, recordings were made, and this was found, and this is reflected in the text. Nevertheless, it has been emphasised in the text.
  2. In relation to the parental role model, we fully agree with the reviewer that it is not only parents who are role models for their children in terms of eating habits, but also those who provide the food. And, in many cases, they suffer from eating disorders or are simply unaware of them. Thank you for the note, which we obviously consider, but perhaps for a further line of research. For example, the article Childhood use of mobile devices: Influence of mothers' socio-educational level (Jiménez-Morales, Montaña & Medina-Bravo - Comunicar, 2020, 28, 64, 21-28) addresses this issue. It is certainly important to study this issue, but we leave it for future lines of research.

  1. In the Spanish case, ethical and legal norms cover these minors with equal treatment. We are aware of this and, in the future, the intention is to do similar experiments making differentiations for each year of age.

  1. The authors rely on the scientific literature to, not identify, but support the concepts of physical health, happiness and well-being. In fact, the complementary literature provided in the explanation of the first question addresses this (Seligman, 2016; Ackerron, 2012; Diener, 2000). In any case, we have followed the reviewer's suggestion and used the expression "subjective happiness".

Sincere thanks for all comments and suggestions for improvement.

Round 2

Reviewer 2 Report

In this second version of the article, the authors have incorporated the comments from the first review. The statistical soundness is low but is the methodology that has been selected by the authors and backed by the bibliography revision.

Author Response

Thank you very much for your comment. 
We acknowledge receipt for future research. 
Yours sincerely. 

Reviewer 4 Report

Thank you for the opportunity to review again the manuscript entitled, "Children, Media and Food. A New Paradigm in Food Advertising, Social Marketing and Happiness Management”. I thank the authors their time and effort. 
My opinion is that in the current form the manuscript does not have scientific quality to be published in IJERPH. The manuscript is below scientific standards.  
The manuscript does not meet my expectations for publication in the journal.
Thank you very much.  

Author Response

Dear Reviewer, 
thank you very much for your time in reading, reviewing and evaluating (and re-evaluating) our study. 
We believe that the methodology used meets the requirements of the social sciences. However, with true appreciation, we proceed to take your considerations into account for future research. 
Thank you very much, again.